# We Need to Talk About Functional Brain Networks

## Abstract

Functional brain networks (fNETs), typically derived from fMRI time series, have been widely studied for understanding demographic differences and neurodegenerative diseases. Recent years have seen an increasing adoption of deep learning methods, particularly graph neural networks (GNNs) and Transformers, for analyzing fNETs. Yet, the structural characteristics of fNETs remain poorly understood, and it is unclear whether these complex architectures consistently outperform simpler baselines. In this work, we conduct a systematic comparison of GNNs and Transformer-based models with baseline models across publicly available fNET datasets. We show that strong baseline models often match or exceed the performance of GNNs, while Transformers demonstrate more consistent gains. Our findings suggest that pooling mechanisms are a potential bottleneck for GNN performance. We argue that careful evaluation with simple baselines is crucial before attributing improvements to architectural sophistication.

## 1   Introduction

Functional brain networks (fNETs) are graph representations of the brain, where nodes correspond to distinct brain regions and edges reflect functional similarities quantified by correlations between their fMRI time series [1]. They have been widely used to study neurodegenerative disorders and demographic differences such as gender [2,3]. Early studies employed handcrafted graph measures and graph kernels to analyze these networks [4,5], but more recent works increasingly rely on deep learning models such as graph neural networks (GNNs) and Transformers [6,7]. Despite their promise, fNETs differ fundamentally from typical graph domains such as molecules or social networks: nodes are fixed and edge weights are dense. Moreover, fNETs lack node features, and connectivity values are also used as node attributes. Yet complex architectures like GNNs and Transformers are frequently applied without systematic baseline comparisons, even though recent studies have shown that, contrary to common belief, simple multilayer perceptrons (MLPs) can outperform GNNs in certain tasks [8,9].

Another major challenge is data availability. Most open datasets release raw fMRI scans rather than processed networks. While standardized preprocessing pipelines improve consistency, openly available preprocessed networks remain scarce. Such resources would enable more consistent benchmarking and reproducible comparisons across studies, but only few open fNET datasets exist.

This work asks two main questions: (1) Are GNNs or Transformers consistently necessary for modeling fNETs? (2) How do pooling strategies affect their performance? To address these questions, we compare GNNs and Transformers with varying pooling strategies against two baselines: a flattened MLP that treats the network as a vector, and a DeepSet model that processes nodes independently using a shared MLP. We argue that any proposed architecture should surpass such baselines. Our experiments across three datasets, two open fNET datasets (ABIDE and HCP) and one private cohort, systematically evaluate whether increased architectural complexity leads to consistent performance gains.

Submitted to 39th Conference on Neural Information Processing Systems (NeurIPS 2025). Do not distribute.

## 2   Datasets

We focus on two widely used open datasets that provide precomputed functional brain networks rather than raw fMRI signals, along with one private dataset.

- **ABIDE:** fNETs were extracted from preprocessed fMRI data provided by the Preprocessed Connectomes Project using Nilearn in Python [10]. Low-quality scans failing quality checks were excluded, leaving 871 scans (403 ASD patients, 468 healthy controls).

- **HCP-Gender:** fNETs released by [11] from the HCP1200 dataset were used for gender classification, including 1,078 subjects labeled as male or female.

- **XXX:** A private dataset consisting of 42 subjects—18 diagnosed with Alzheimer's Disease and 24 with Subjective Cognitive Impairment (SCI). Preprocessing details are provided here.[1]

## 3   Methodology

We denote the fMRI time series of a brain region $n$ as $x_n \in \mathbb{R}^T$ where $T$ is the number of time points. With $N$ brain regions in total, the functional network (fNET) can be represented as a graph $\mathcal{G} = (\mathcal{V}, \mathcal{E}, A)$. The vertex set $\mathcal{V}$ corresponds to brain regions, and each edge in the edge set, $\mathcal{E}$, is given by the correlation coefficient between regional time series: $e_{i,j} = \mathrm{corr}(x_i, x_j)$. and $A$ is the adjacency matrix that stores edges, $A_{i,j} = e_{i,j}$

We consider five architectures: 1) **GCN** [12] : $f(A, X) = \sigma(\tilde{L}X\Theta)$, with $\tilde{L}$ is normalized Laplacian of adjacency matrix, $X$ is the node feature matrix, $\Theta$ trainable parameters and $\sigma$ is a nonlinear function, 2) **GAT** [13] : $f(A, X) = \sigma(\tilde{A}X\Theta)$, where $\tilde{A}$ attention coefficient matrix, learned only for connected nodes specified by $A$, 3) **Transformer** [14] : $f(X) = \sigma(\mathrm{self\_attn}(X))$, allowing every node to attend to all others by applying conventional self-attention mechanism, 4) **MLP:** flattens the fNET and computes $f(X) = \mathrm{MLP}(\mathrm{vec}(X))$, 5) **DeepSet** [15] : ignores adjacency and computes $f(X) = \mathrm{MLP}(X)$.

These models produce latent node representations, $\hat{X}$, which are aggregated into a graph-level representation using pooling. We consider three schemes: 1) **Basic Pooling:** sum or mean of node features, $z = \sum_n \hat{x}_n$ or $z = \frac{1}{N}\sum_n \hat{x}_n$. 2) **Concat Pooling:** concatenation of node features followed by a linear projection, $z = W[\hat{x}_1; \hat{x}_2; \ldots; \hat{x}_N]$. 3) **Soft Pooling:** nodes are assigned to $K$ orthogonal clusters via $S \in \mathcal{R}^{N \times k}$ with pooled features $Z = S^T\hat{X}$. Resulting $Z$ is then processed like Concat Pooling [7].

## 4   Experiments and results

We compared GCN, GAT, and Transformers with various pooling strategies against simpler baselines (MLP and DeepSet) under consistent training conditions. For ABIDE and HCP-Gender, we use a (0.7:0.1:0.2) train/validation/test split, repeat experiments 5 times, and report mean and standard deviation of accuracy, F1, and AUC. The best model is selected on the validation set based on AUC, and its performance is reported on the test set. We tuned hyperparameters based on validation AUC and report the test performance of the best configuration. Models were trained for up to 100 epochs using the Adam optimizer, exploring learning rates $\{1e-3, 1e-4\}$, weight decays $\{0, 1e-3, 1e-4\}$, layers $\{1, 2, 3\}$ and, hidden dimension $\{8, 64\}$. For the XXXX dataset, we perform leave-one-out cross-validation due to its small size. Hyperparameters are set equal to those in the ABIDE and HCP-Gender experiments. To make GNN models suitable, we apply percentile-based thresholding, retaining the top 5 percent of edges in the adjacency matrix. We used adjacency matrix as the node feature matrix as it is a common practice in the field [16]. All experiments were implemented in Python and run on a single NVIDIA RTX 3060. The code used for these experiments is publicly available here[2]. We report the results for different models in Table 1. The main observations are as follows :

---

[1]Details of the preprocessing pipeline are not shared yet due to anonymity.

[2]Github repo will be shared upon acceptance

Table 1: Performance of different architectures and pooling schemes on three datasets (mean ± std). Top results are highlighted: best in red, second-best in blue. The number of nodes in each graph is indicated by the dataset name: ABIDE and XXXX use the Schaefer-400 atlas, while HCP-GENDER uses the Schaefer-1000 atlas to define brain regions [17].

| Model | Pooling | ABIDE(400) | | | HCP-Gender(1000) | | | XXXX(400) | | |
|---|---|---|---|---|---|---|---|---|---|---|
| | | Acc | F1 | AUC | Acc | F1 | AUC | Acc | F1 | AUC |
| GCN | Mean | 0.667±0.023 | 0.630±0.043 | 0.744±0.015 | 0.806±0.020 | 0.804±0.019 | 0.897±0.006 | 0.700±0.011 | 0.627±0.018 | 0.762±0.008 |
| | Sum | 0.633±0.034 | 0.622±0.054 | 0.690±0.031 | 0.751±0.009 | 0.750±0.008 | 0.821±0.024 | 0.600±0.046 | 0.548±0.044 | 0.631±0.037 |
| | Concat | 0.706±0.020 | 0.674±0.053 | 0.774±0.019 | 0.839±0.012 | 0.838±0.012 | 0.911±0.014 | 0.728±0.035 | 0.653±0.053 | 0.803±0.020 |
| | Soft | 0.687±0.022 | 0.614±0.056 | 0.763±0.007 | 0.848±0.010 | 0.847±0.010 | 0.914±0.008 | 0.729±0.035 | 0.679±0.032 | 0.808±0.018 |
| GAT | Mean | 0.685±0.028 | 0.657±0.049 | 0.738±0.017 | 0.825±0.016 | 0.824±0.016 | 0.896±0.006 | 0.633±0.028 | 0.554±0.030 | 0.712±0.026 |
| | Sum | 0.652±0.028 | 0.614±0.074 | 0.689±0.029 | 0.719±0.020 | 0.718±0.020 | 0.790±0.025 | 0.629±0.038 | 0.639±0.048 | 0.558±0.072 |
| | Concat | 0.713±0.019 | 0.665±0.028 | 0.771±0.011 | 0.822±0.014 | 0.821±0.014 | 0.896±0.008 | 0.652±0.032 | 0.572±0.022 | 0.717±0.045 |
| | Soft | 0.696±0.035 | 0.621±0.064 | 0.762±0.011 | 0.808±0.027 | 0.807±0.028 | 0.894±0.020 | 0.652±0.019 | 0.574±0.053 | 0.746±0.023 |
| Transformer | Mean | 0.691±0.016 | 0.660±0.022 | 0.765±0.009 | 0.835±0.023 | 0.834±0.023 | 0.919±0.019 | 0.705±0.024 | 0.634±0.029 | 0.754±0.026 |
| | Sum | 0.654±0.018 | 0.604±0.055 | 0.697±0.019 | 0.822±0.014 | 0.821±0.014 | 0.890±0.016 | 0.638±0.055 | 0.575±0.072 | 0.663±0.044 |
| | Concat | 0.724±0.034 | 0.637±0.092 | 0.814±0.033 | 0.888±0.017 | 0.887±0.018 | 0.961±0.007 | 0.747±0.024 | 0.697±0.031 | 0.784±0.039 |
| | Soft | 0.719±0.042 | 0.683±0.033 | 0.803±0.037 | 0.810±0.143 | 0.771±0.216 | 0.862±0.183 | 0.785±0.015 | 0.734±0.019 | 0.822±0.006 |
| MLP | - | 0.725±0.010 | 0.657±0.017 | 0.808±0.002 | 0.906±0.003 | 0.906±0.003 | 0.962±0.001 | 0.805±0.009 | 0.757±0.015 | 0.827±0.003 |
| DeepSet | Mean | 0.688±0.020 | 0.656±0.027 | 0.757±0.008 | 0.823±0.011 | 0.821±0.011 | 0.893±0.004 | 0.681±0.011 | 0.573±0.008 | 0.726±0.009 |
| | Sum | 0.667±0.022 | 0.615±0.057 | 0.722±0.014 | 0.817±0.023 | 0.815±0.023 | 0.902±0.012 | 0.681±0.041 | 0.622±0.044 | 0.726±0.022 |
| | Concat | 0.734±0.012 | 0.691±0.023 | 0.806±0.011 | 0.860±0.015 | 0.859±0.015 | 0.931±0.009 | 0.705±0.024 | 0.634±0.042 | 0.769±0.017 |
| | Soft | 0.721±0.014 | 0.646±0.021 | 0.800±0.019 | 0.867±0.009 | 0.866±0.009 | 0.940±0.007 | 0.724±0.024 | 0.666±0.038 | 0.792±0.014 |

**GNNs and Transformer do not outperform baseline models.** Although Transformer with concat or soft pooling achieve performance comparable to the baselines in some cases, they do not introduce significant gains. MLPs consistently perform better than other models across most tasks and metrics.

**GNNs fail to outperform the graph-free DeepSet model.** This suggests that the creating sparse underlying graph from fNETs is difficult to define reliably and requires further investigation.

**Pooling strategies have a critical impact on performance.** Simple pooling methods underperform, while concatenation-based pooling generally yields better results, indicating that inadequate pooling may be a key bottleneck in fNET analysis.

Our results highlight the importance of strong baseline models for demonstrating genuine performance improvements. These observations are consistent with prior studies [18,19]. For instance, it has been shown that a simple MLP applied directly to time-series data can outperform Transformer-based models [18]. Although their focus was on time-series signals rather than brain networks, the study highlights the value of robust baselines. Similarly, other work has reported that simple models can surpass more complex architectures [19]. Our main contribution is to extend these observations to fully open fNET datasets and to emphasize the critical influence of pooling strategies.

# 5 Conclusion

Our experiments demonstrate that simple baseline models, such as MLPs and DeepSet, can outperform complex architectures like GNNs and Transformers on functional brain network analysis. These results highlight the critical importance of carefully evaluating model design choices, particularly graph pooling strategies, before claiming performance improvements. By conducting systematic comparisons on fully open network datasets, we validate that strong baselines are essential for reproducible and fair benchmarking. Our study emphasizes that future works should report baseline performances and carefully consider pooling mechanisms to meaningfully demonstrate the benefits of more sophisticated architectures. Furthermore, recent studies questioning the necessity of GNNs should be followed closely by researchers. Rather than proposing increasingly complex architectures solely to achieve marginal performance gains, focusing on interpretability and understanding the learned representations may offer a more valuable direction for advancing functional brain network analysis.

# Broader Impact

Our work focuses on analyzing functional brain networks using machine learning. Potential positive impacts include improved understanding of neurodegenerative disorders and supporting research in neuroscience and clinical decision-making. Potential negative impacts may arise if the models are misused for clinical predictions without proper validation, potentially leading to incorrect diagnoses. These models are intended solely for research purposes and should not be used for direct clinical decision-making.

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
