# OpenReview forum: "We Need to Talk About Functional Brain Networks"
_EurIPS.cc/2025/Workshop/MedEurIPS — EurIPS 2025 Workshop MedEurIPS Submission_

### Official Review · Reviewer_NLpQ · 2025-10-24
**Questioning the SOTA in fMRI analysis with good evaluation.**

**Rating:** 8
**Confidence:** 4

**Review:**

Strengths:
-	Important systematic comparison
-	Novel findings questioning state-of-the-art approaches, which might stimulate useful discussion
-	solid evaluation
Weaknesses:
-	XXX / XXXX is a questionable name for a private dataset.
-	Reporting standard error instead of std would make it easier to assess the statistical validity

---

### Official Review · Reviewer_gDyT · 2025-10-31
**Nice work that fits the theme**

**Rating:** 8
**Confidence:** 4

**Review:**

This work is about functional brain networks, the methods that used for analysing it and future directions. I think this will bring interesting discussions, especially about how to use probabilistic tools for specific medical task.

---

### Decision · Program_Chairs · 2025-10-31

**Decision:**

Accept (Oral)

**Comment:**

Both reviewers praise the clarity and relevance of this paper, which provides a systematic comparison of GNNs, Transformers, and simpler baselines for functional brain network analysis.